# Hafnium—an optical hydrogen sensor spanning six orders in pressure

C. Boelsma[1], L.J. Bannenberg[2], M.J. van Setten[3], N.-J. Steinke[4], A.A. van Well[2] & B. Dam[1]

Hydrogen detection is essential for its implementation as an energy vector. So far, palladium is considered to be the most effective hydrogen sensing material. Here we show that palladium-capped hafnium thin films show a highly reproducible change in optical transmission in response to a hydrogen exposure ranging over six orders of magnitude in pressure. The optical signal is hysteresis-free within this range, which includes a transition between two structural phases. A temperature change results in a uniform shift of the optical signal. This, to our knowledge unique, feature facilitates the sensor calibration and suggests a constant hydrogenation enthalpy. In addition, it suggests an anomalously steep increase of the entropy with the hydrogen/metal ratio that cannot be explained on the basis of a classical solid solution model. The optical behaviour as a function of its hydrogen content makes hafnium well-suited for use as a hydrogen detection material.

[1] Faculty of Applied Sciences, Department of Chemical Engineering, Delft University of Technology, Van der Maasweg 9, 2629 HZ Delft, The Netherlands. [2] Faculty of Applied Sciences, Department of Radiation Science and Technology, Delft University of Technology, Mekelweg 15, 2629 JB Delft, The Netherlands. [3] Department of Nanoscopic Physics, Institute of Condensed Matter and Nanosciences, Catholic University of Leuven, Chemin des Toiles 8, 1348 Louvain-la-Neuve, Belgium. [4] ISIS Neutron and Muon Source, Rutherford Appleton Laboratory, OX11 0QX Oxford, UK. Correspondence and requests for materials should be addressed to B.D. (email: b.dam@tudelft.nl).

Monitoring the partial pressure of hydrogen in a cheap and reliable way is essential for its implementation as an energy vector. Optical sensors have a particular safety benefit due to the lack of electric wires in the area of operation due to the remote readout[1,2]. The change in optical properties of a metal hydride when exposed to hydrogen make them particularly suited for this purpose. These changes result in a change in optical reflection and a shift in the surface plasmon resonance absorption peak[1,3,4]. In a micro-mirror configuration where the film is deposited on top of an optical fibre[5–9], the change in optical reflection can be used to measure the ambient hydrogen pressure.

So far, palladium is considered to be the most effective hydrogen-sensing material[10–12]. Typically, at room temperature this material detects hydrogen between partial $H_2$ pressures of $10^{+1}$–$10^{+4}$ Pa with a response time of less than a minute. However, the sensor-to-sensor reproducibility is limited. In addition, these sensors suffer from blistering/delamination and micro-cracking[13,14]. The most important drawback of $PdH_x$ is its hysteresis: the optical response to an increasing hydrogen pressure differs from that of a decreasing hydrogen pressure[13,15,16]. This is a result of the first order phase transition to the ordered $PdH_x$ β-phase. This phase transition is suppressed by alloying, which results in a strongly reduced optical signal compared with pure $PdH_x$, whereas the pressure range is shifted towards higher pressures[13,14,17,18].

Here we present Pd-capped hafnium as an alternative optical hydrogen sensing material. The Pd cap-layer takes care of the dissociative absorption of hydrogen and the presence of hydrogen is monitored by the change in optical properties of $HfH_x$. This allows us to measure the hydrogen pressure over more than six orders in magnitude. The sensing properties of this material show a high reproducibility and no signs of hysteresis. Moreover, the temperature dependence of the optical output appears to be highly linear, facilitating the calibration of such a sensor. Although the thermodynamics of this material are not yet fully understood, our results suggest a path for tailored design of a wider range of transition metal based optical hydrogen sensors.

## Results

**Phase behaviour of $HfH_x$.** To understand the optical behaviour and hydrogen-sensing properties of $HfH_x$ thin films, we first discuss their structural behaviour.

In the bulk phase diagram (see Fig. 1), a large two-phase region separates the α hexagonal-close-packed (hcp) solid solution from the δ face-centred-cubic (fcc) phase, where hydrogen occupies the interstitial tetrahedral sites[19–21]. Upon further hydrogenation, a transformation to the ε face-centred tetragonal (fct) phase is observed, in which the fcc lattice is compressed along the c axis. The fcc and fct phases coexist between $1.78 < x < 1.86$. In addition, at around $x = 1.5$, a metastable, deformed cubic $δ'$ phase—which disappears above 80 °C—was reported[21]. The volume of the host metal increases on hydrogenation: 16% on entering the fcc phase and an additional 2.8% within the fcc phase[19]. Remarkably, the transformation to fct leads to a volume reduction of 1.2%, which is only partially recovered on saturation. As compared with the hcp phase, the saturated fct phase is expanded 18.1% in volume (see Supplementary Table 1).

Reports on the analysis of the enthalpy and entropy of hydrogenation are not very consistent. From the review by Mintz[19], it follows that the enthalpy of formation of the hcp/fcc transition is almost the same as for the fcc/fct transition. Similarly, the Sieverts data from Luo et al.[22] show an almost constant enthalpy of formation for $0.5 < x < 2$, albeit with much larger enthalpy values ($-112$ versus $-142$ kJ (mol $H_2$)$^{-1}$). Following the analysis of Mintz[19], at the hcp/fcc boundary,

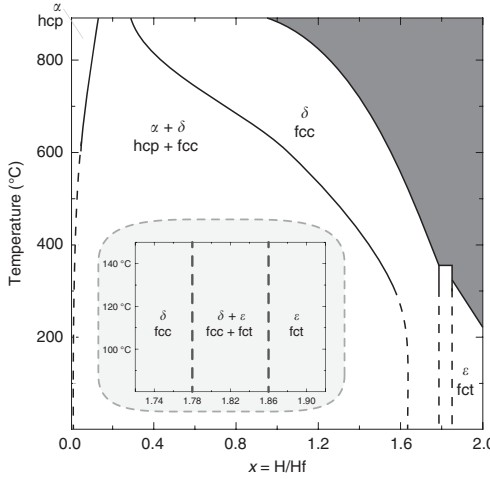

**Figure 1 | $HfH_x$ phase diagram.** Phase diagram of bulk $HfH_x$ adapted after Mintz[19] showing a hcp–fcc coexistence at low x; a solubility range in the fcc phase ranging from $x = 1.68$ to 1.78, whereas for $x > 1.86$ a fct phase is observed. The shaded area indicates the region where the hydride formation pressure exceeds 1 atm. The inset shows the range between $1.72 \leq x \leq 1.92$ in more detail.

the entropy equals 83.2 J (K mol $H_2$)$^{-1}$ and rises to 173.5 J (K mol $H_2$)$^{-1}$ at the fcc/fct phase boundary.

In sputtered Hf thin films, the bulk phase behaviour—which is obtained mostly at high temperatures—is reproduced qualitatively. Using neutron reflectometry (NR) and X-ray diffraction (XRD; see Methods section) we identify three states: the as-prepared state, the saturated state and the unloaded state. The as-prepared state corresponds to the situation directly after sputtering, without exposing the sample yet to hydrogen ($x = 0$). XRD confirms that these as-deposited thin films have the hcp structure with similar lattice constants as those reported for bulk (see Supplementary Fig. 1 and Supplementary Table 2). The saturated state is obtained by exposing the sample to a partial hydrogen pressure of $10^{+3}$ Pa at 120 °C. From in situ NR at 12 °C (see Fig. 3) we find a hydrogen content $x = 1.98 \pm 0.02$, similar to the saturated bulk composition in the fct phase[21].

The unloaded state corresponds to the situation after exposure of a hydrogenated sample to air for (at least) 3 days. The film does not return to the as-prepared hcp phase. Instead, it remains in the fcc phase, obtaining almost the same lattice parameters as reported for bulk $HfH_{1.62}$. Even after an exposure to air at room temperature over more than 6 months, the film remains in the fcc phase without any change in the XRD diffraction pattern (see Supplementary Fig. 1). The exact position of the phase boundary at low temperature is disputed in the literature. The extrapolation of the high temperature data ($> 300$ °C) suggests an fcc phase boundary between $1.5 < x < 1.62$ (refs 19,22). Using NR we find values around $x \cong 1.43 \pm 0.02$ for fully dehydrogenated thin films. The fact that the material does not revert to the hcp phase is similar to the Y–H system, where upon dehydrogenation, $YH_3$ transforms to $YH_{1.9}$ and remains in this phase even after (a lengthy) exposure to oxygen[2].

To observe the nature of the hcp–fcc phase transition, we apply a partial hydrogen pressure of 0.5 Pa at 120 °C for 6 h using in situ XRD. We find that the hcp and fcc phases coexist during the phase transition. We observe a decrease of the (002) Hf hcp peak intensity while at the same time the (111) $HfH_x$ fcc peak intensity increases (see Fig. 2). The lattice reflections of both phases remain the same upon hydrogenation. This means that, as in bulk, the hcp to fcc phase transition in thin films is first-order and proceeds by the co-existence of two (incoherent) phases.

Such a two-phase region is not observed at the fcc→fct phase transition. Starting from the unloaded state we observe that the (111) reflection remains fairly constant up to 5 Pa, above which the 2θ peak position starts to shift to lower angles (see Fig. 2b). This corresponds to an increase of the spacing of the (111) fcc lattice planes. Above 30 Pa the 2θ peak position increases again,

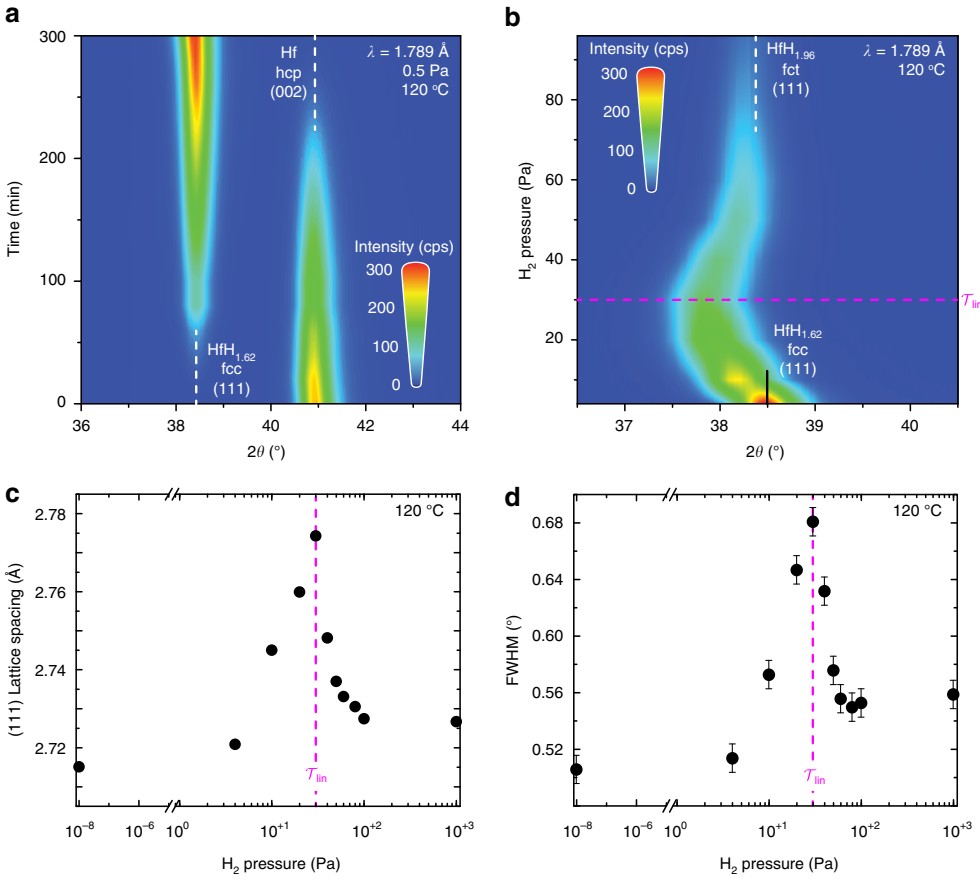

**Figure 2 | HfH$_x$ phase transitions.** Contour plots of consecutive *in situ* XRD patterns of HfH$_x$ upon hydrogenation, corresponding to (**a**) the hcp–fcc phase transition (studied by exposing a 40 nm Hf film to 0.5 Pa at 120 °C) and (**b**) the fcc–fct phase transition (obtained by increasing the partial hydrogen pressure from 4 to 100 Pa at 120 °C). Here, red indicates a high and blue a low intensity. It is noteworthy that the decrease in intensity with hydrogen pressure is the result of an increase shielding of the X-rays due to the increase of Argon gas in the cell. In both plots, the (dashed) lines indicate the peak positions of hcp Hf, fcc HfH$_{1.62}$ and fct HfH$_{1.98}$ as reported for bulk[21]. (**c**) The corresponding (111) d-spacing and (**d**) the full-width-half-maximum FWHM as a function of the applied hydrogen pressure obtained from the XRD patterns. It is noteworthy that the pressure axis in **c,d** has a break between $1.1 \times 10^{-6}$ and 0.9 Pa. The magenta dashed lines in **b–d** indicate the optical state $\mathcal{T}_{lin}$, corresponding to a pressure of 30 Pa at 120 °C. The error bars in **c,d** equal the s.d. of the fitted peak positions using the patterns shown in **a,b**.

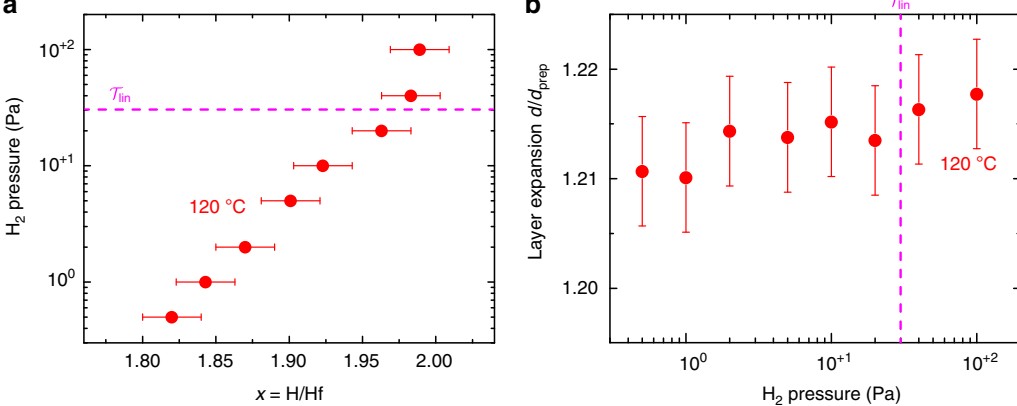

**Figure 3 | Thickness and H/Hf ratio as derived from *in situ* NR.** (**a**) The applied hydrogen pressure plotted as a function of the hydrogen concentration as derived from *in situ* NR. (**b**) The thin-film thickness normalized to the as-prepared film thickness plotted as a function of pressure. The NR measurements are performed on a Pd-capped Hf thin film (40 nm) at 120 °C (see Methods). The magenta dashed indicate the optical state $\mathcal{T}_{lin}$, corresponding to a pressure of 30 Pa at 120 °C. The error bars follows from the fitting procedure (see methods).

resulting in a decreased *d*-spacing (see Fig. 2c). Above 60 Pa the $2\theta$ peak position hardly changes and almost matches the (111) fct $2\theta$ peak position reported for bulk $HfH_{1.98}$ (ref. 21). The smooth transition of the $2\theta$ peak position from the unloaded fcc phase to the saturated $HfH_{1.98}$ fct phase indicates the presence of a coherent transition wherein the two phases are elastically coupled. This is supported by the observation that the width of the diffraction peaks widens during the transition (see Fig. 2d). The fact that the fcc (111) lattice spacing gets larger than that of fct at maximum pressure, suggests a deformation of the cubic lattice, as a short fct axis should develop along the [001] direction. Unfortunately, the low peak intensities preclude an analysis of the out-of-plane diffraction peaks to further explore the symmetry of the film during the phase transition.

Using *in situ* NR, we measured the film thickness and the hydrogen content at various stages of hydrogenation. The thickness appears to be constant or slightly decreasing at hydrogen pressures between 0.5 and 30 Pa (see Fig. 3). This contrasts with the prominent changes in the (111) lattice spacing as measured by XRD above 5 Pa. Similarly, we find a largely linear relation between the hydrogen content and $\ln(P/P_0)$. At 30 Pa we calculate a hydrogen content $x \cong 1.96$, far above the critical hydrogen content reported for the bulk fcc/fct transition $(1.78 < x < 1.86)^{19}$. Above 30 Pa we observe a deviation from linearity resulting in a saturated hydrogen content of $x \cong 1.98$. In metal hydrides the volume expansion of a thin film on hydrogenation is usually fully converted in a thickness increase. Comparing the as-prepared with the fully saturated state, we find that the Hf thickness expands by $21.5 \pm 0.5\%$. This is somewhat larger than the volume expansion measured in bulk. Possibly, this is due to an incomplete transition to the fct phase. On dehydrogenation in air, we find that the expansion reduces to around 16%, comparable to the bulk volume expansion. At that stage the hydrogen content of the film measures $x \cong 1.43$.

**Optical behaviour of HfH$_x$.** By means of hydrogenography[13,16,23], we monitored the (white light) optical transmission $\mathcal{T}$ of a thin film as a function of the partial hydrogen pressure $P$ between $10^{-3}$–$10^{+5}$ Pa, at 90, 120 and 150 °C. Figure 4 shows the optical response of a 40 nm-thick as-prepared Hf film capped by 10 nm Pd to a constant partial hydrogen pressure of 0.4 Pa at 120 °C. In this figure we express the transmission change with respect to the as-prepared state $\mathcal{T}_{prep}$. The optical transmission first increases and then decreases to a stable level. Combining this with the structural results we find that the increase of the film transmission is due to the hpc→fcc transition, whereas the decrease of the transmission occurs within the fcc phase. When exposing the hydrogenated film first to vacuum ($10^{-4}$ Pa) and then to oxygen for several days, we find that the transmission returns to $\ln(\mathcal{T}/\mathcal{T}_{prep}) = +0.187$. Remark that from our XRD and NR data, we deduce that this level corresponds to the unloaded fcc state at $x \cong 1.43$. Optically, we define the corresponding transmission by $\mathcal{T}_{unl}$. In the remainder of this study, we use this level as the new reference point, such that the unloaded state is indicated by $\ln(\mathcal{T}/\mathcal{T}_{unl}) = 0$. This reference point was measured independently for all three temperatures.

Increasing the pressure beyond 0.4 Pa results in a further decrease of the transmission, which saturates at $10^{+3}$ Pa at a transmission level $\ln(\mathcal{T}/\mathcal{T}_{prep}) = -0.218$ with respect to the transmission level of the unloaded state. We define this saturated transmission level by $\mathcal{T}_{sat}$. From the neutron-data (see Fig. 3) we learn that this state corresponds to fct $HfH_{1.98}$. Remarkably, exposing the system to intermediate pressures results in distinct optical levels, which are stable over time, reversible and hysteresis-free (see Fig. 4b). It is noteworthy that the Pd cap layer has no contribution to this optical behaviour as the

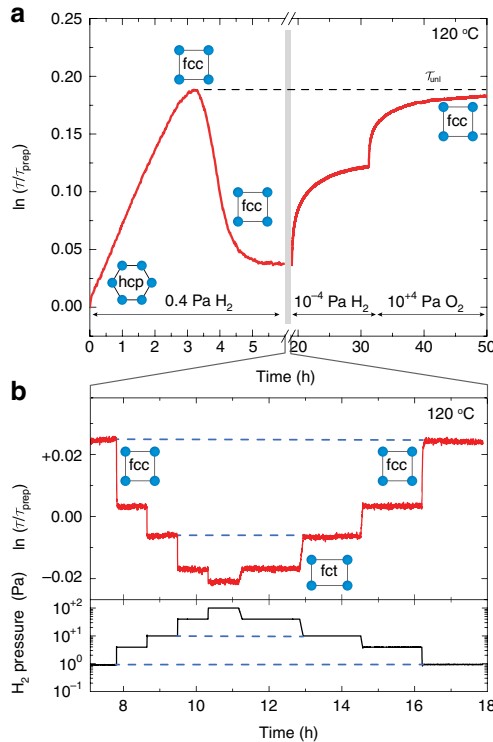

**Figure 4 | Optical response of HfH$_x$ to hydrogen.** (**a**) The (white light) optical transmission change at 120 °C of a Pd-capped Hf film (40 nm) as function of time, where the film is exposed for the first six hours to 0.4 Pa H$_2$ and later for 12 h to vacuum (corresponding to $10^{-4}$ Pa H$_2$) and for 18 h to $10^{+4}$ Pa O$_2$. The dashed line indicates the maximum transmission $\mathcal{T}_{unl}$, reached after (long) exposure to oxygen. (**b**) Close-up of the time period 7–18 h where the partial hydrogen pressure varies in steps between 100 and $10^{+2}$ Pa. Here, the dashed lines mean the levels of same transmission/pressure, indicating the absence of hysteresis.

Pd optical response starts at much higher pressures ($10^{+4}$ Pa at 120 °C)[13,17,18].

In Fig. 5 we plot the optical transmission as a function of hydrogen pressure for various temperatures, resulting in so-called pressure-transmission-isotherms (P$\mathcal{T}$Is). Here, each data point corresponds to the optical transmission reached after exposing the film to a constant pressure between $10^{-3}$ and $10^{+4}$ Pa for an hour. A large part of the P$\mathcal{T}$I is determined by a linear relation between $\ln(\mathcal{T}/\mathcal{T}_{unl})$ and the applied hydrogen pressure $\ln(P/P_0)$. Comparing the optical data with the NR results, we conclude that $\ln(\mathcal{T}/\mathcal{T}_{unl})$ is proportional to the hydrogen content for $1.8 < x < 1.98$ (Supplementary Fig. 2). This behaviour is, to our knowledge, not unique, as a similar relation is found in, for example, VH$_x$ (refs 24,25). It suggests that the proportionality between $\ln(\mathcal{T}/\mathcal{T}_{unl})$ and $x$ is also valid for $x < 1.8$. Indeed, the extrapolation of the hydrogen fraction to $\mathcal{T} \to \mathcal{T}_{unl}$ suggests $x \cong 1.42$. This equals the hydrogen content measured in the dehydrogenated films: $x \cong 1.43$. The extrapolation also implies that 120 °C a hydrogen pressures as low as $10^{-7}$ Pa can be optically detected. At present we are unable to verify this extrapolation. Although the linear relation between hydrogen content, $\ln(P/P_0)$ and the optical transmission may get lost at lower pressures, thermodynamic bulk data corroborates our extrapolation of the hydrogen equilibrium pressure. At the fully desorbed fcc state, it should correspond to the hcp/fcc plateau pressure. When we calculate the plateau pressures from the bulk thermodynamic data[19] for the three temperatures, we find values for the equilibrium pressure (closed symbols in Fig. 5) closely

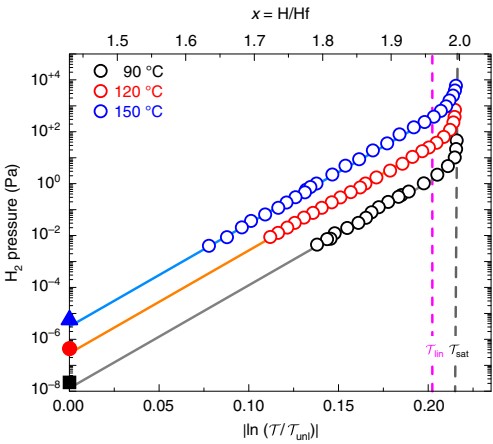

**Figure 5 | Optical transmission isotherms of HfH$_x$.** Pressure-transmission-isotherms (P$\mathcal{T}$Is) measured at 90 °C (black squares), 120 °C (red circles), and 150 °C (blue triangles) of a Pd-capped Hf (40 nm) film, after reaching the unloaded state represented by $\mathcal{T}_{unl}$ (that is, $\ln(\mathcal{T}/\mathcal{T}_{unl})=0$). The open symbols are thin film data measured by hydrogenography. The solid lines are extrapolations of the measured data. The closed symbols are the hcp/fcc plateau pressures, calculated from the bulk thermodynamic data[19]. The dashed vertical line indicates the saturation transmission $\mathcal{T}_{sat}$. The hydrogen fraction $x$ in HfH$_x$ is based on the NR data obtained at 120 °C. The 120 °C data are the average of multiple (sub)cycles, whereas the 90 °C and 150 °C data are obtained from a single cycle. The magenta dashed line indicates the optical state $\mathcal{T}_{lin}$, corresponding to a pressure of 30 Pa at 120 °C. The lower limit of $10^{-3}$ Pa is determined by our experimental setup.

matching the extrapolated thin film data. Hence, although at 150 °C the measurement ranges over six orders in pressure, the extrapolation suggests that even a sensor ranging over ten orders in pressure might be feasible.

At 120 °C we observe that for pressures above 30 Pa the linear behaviour is limited by the upward trend in pressure close to the saturated optical signal. This correlates with the XRD data where the (111) lattice spacing decreases above ~30 Pa. Remarkably, we do not see any optical effect at the onset of the fcc/fct transition at 5 Pa: no change in slope or kink in the optical transmission is observed. This may be due to the coherent nature of the phase transition. In addition, DFT calculations show that the optical properties are hardly affected by the phase transition (see Supplementary Methods and Supplementary Fig. 3a). Hence, the phase transition will only have a very small optical effect.

When adding a H-vacancy to the fcc HfH$_2$ structure, we obtain a computational model for the HfH$_{1.5}$ phase. Moving from HfH$_{1.5}$ to HfH$_2$, we observe a shift of spectral weight in the region below 3 eV below the Fermi level. Excitations involving these states are too high in energy to affect the visible spectrum. On the other hand, we observe an increase in spectral weight in the first two eV above the Fermi level. This leads to an increase in the joined density of state and hence stronger absorption in the optical part of the spectrum. It is noted that in both cases the intra-band plasma frequency lies well beyond the optical spectrum.

The temperature behaviour of the P$\mathcal{T}$Is is quite unusual and not easy to understand within the context of a classical thermodynamic description of metal hydrides. The P$\mathcal{T}$Is shift uniformly to higher pressures when increasing the temperature, leading to a single master curve when normalized (see Supplementary Fig. 2). According to the Van't Hoff analysis (see Supplementary Methods) as visualized in Supplementary Fig. 4a, this implies that the enthalpy of dissolution does not depend on the hydrogen content. For sensor applications, this behaviour is highly beneficial, as it facilitates the temperature

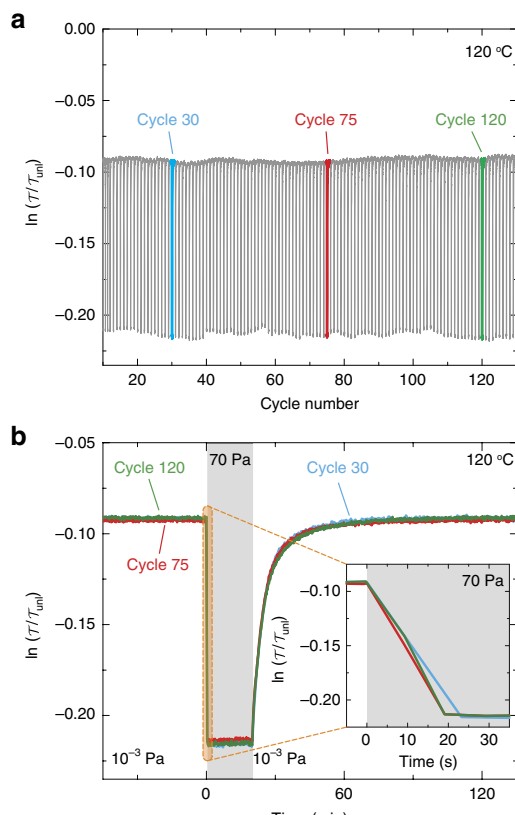

**Figure 6 | Sensing properties of HfH$_x$.** (**a**) Optical response of the Pd-capped Hf film to 130 (sub)cycles at 120 °C varying the pressure between $10^{-3}$ and 70 Pa H$_2$. (**b**) Time dependence of three individual hydrogenation cycles showing the high reproducibility of the kinetics.

calibration: to compensate for a change in temperature the same constant value needs to be added to obtain $\ln(P/P_0)$. Assuming the Van't Hoff reasoning to be valid, the linear relation between the pressure and the hydrogen content implies that the entropy of the hydrogen dissolution reaction increases linearly with $x$ (see Supplementary Fig. 4b), albeit over a seemingly unphysically large range of entropy values $-240 < \Delta S < -180$ J (K mol H$_2$)$^{-1}$.

For bulk, it was reported that HfH$_x$ obeys the relation $\log[\sqrt{P/P_0}] \sim \log[x/(1+x)]$ in the fcc phase[19,21]. As shown in Supplementary Fig. 5, the same bulk data yield a linear behaviour when plotting as $\ln(P/P_0) \sim x$. However, the bulk is markedly different in the sense that a kink is observed at the fcc/fct transition, which is clearly absent in thin films. We conclude that in thin films the fcc behaviour is extended over a larger range, due to the coherent nature of the transition. Please note that we do not obtain a decent fit when plotting the thin film data as $\log[\sqrt{P/P_0}] \sim \log[x/(1+x)]$.

**Hydrogenation kinetics.** The cycle stability of the hafnium films is excellent. Figure 6a shows the optical response to 130 cycles of complete hydrogenation (70 Pa) and partially dehydrogenation ($10^{-3}$ Pa) cycles at 120 °C. Apart from some small oscillations in the minimum and maximum transmittance due to fluctuations in the light source, the cycles are identical (see Fig. 6b) and no signs of degradation are observed. In addition, structurally no degradation nor delamination is observed. Even after a 6 months exposure to (open) air, the film still responds in a similar way. Only a reduction of the kinetics is observed, which is likely to be due to a contamination of the Pd-surface.

The hydrogenation kinetics is an important parameter to determine the applicability of Hf as a wide range hydrogen sensor. As shown in Fig. 6b, the films respond reproducibly within 20 s to a pressure increase from $10^{-3}$ to 70 Pa. Here we define the switching time as the time needed to respond to 90% of the saturation level. As shown in Supplementary Fig. 6, also small ($<10^{-2}$ Pa) pressure steps result in an easily observable optical change within 10 s. As a result of the fast switching, the transmission nicely follows positive and negative pressure steps over time in Fig. 7a. This suggests that the sensor speed is mainly diffusion controlled. According to diffusivity data from Levitin et al.[26], a 40 nm film should hydrogenate from its unloaded state within 4–20 s at 120 °C. However, the desorption is much ($\sim$factor 10) slower than the absorption (see Fig. 7a). At low desorption pressures, this is more evident and even a delay in response is observed (see Fig. 7b). The slow desorption is a common feature observed in metal hydride thin films. It is partly due to the difference in chemical potential of the hydrogen in the sensing metal as compared to the Pd cap layer[27]. In addition, in non-ultra-high-vacuum conditions, surface poisoning by adsorbed species ($H_2O$) may play a role[28].

At room temperature the kinetics is much slower than expected from the diffusion data. As shown in Supplementary Fig. 7, at room temperature it takes 3.75 h to obtain 90% of the saturated optical level when applying a pressure of 1 Pa. This is probably due to surface catalytic effects becoming dominant. After coating the Pd cap-layer by a thin (30 nm) layer of polytetrafluorethene (PTFE), the switching time is reduced to only 30 min. This behaviour is in agreement with our earlier studies, which show that the dissociative absorption of hydrogen at room temperature can be improved by surface coatings[28–30]. Apparently, the PTFE

coating keeps Pd surface sites available that would otherwise be blocked by adsorbents such as $H_2O$.

The experimental error in transmission is $\delta\mathcal{T}/\mathcal{T}=10^{-3}$. As a result the accuracy in determining the hydrogen pressure equals $\delta P/P\cong10^{-1}$. Please note that this sensitivity is based on the use of a white light source. A further optimization is possible by selecting the appropriate wavelength. For example, the optical contrast can be tripled using a 635 nm light source (see Supplementary Fig. 8).

## Discussion

Thermodynamically, the most important feature we identified in the Hf films is the linear relation between the hydrogen pressure $\ln(P/P_0)$ and the concentration $x$. The range of $x$ is too large to be fitted to a classical Sieverts behaviour ($x$ linear in $\sqrt{P/P_0}$). Furthermore, we find that the $P\mathcal{T}$Is scale with temperature. As a result, a Van't Hoff analysis (see Supplementary Methods) yields a constant enthalpy, whereas the entropy varies and reaches large, seemingly unphysical values.

The constant enthalpy is remarkable, as we would expect to observe elastic, electronic and thermal contributions[31,32]. Given the lattice expansion from 16% at the fcc phase boundary to 21% at saturation, we expect an attractive elastic H–H interaction to contribute to the enthalpy. It could be partially compensated by an electronic term, but this is unlikely, as the density of states (DOS) at the Fermi level does not change much on hydrogenation (see Supplementary Fig. 3b). The effect of temperature on the electronic term is usually negligible[32]. Possibly, the heat capacity $C_p$ may contain a term linear in $x$, which would yield an $xT$ term in the enthalpy. In this way, an enthalpic term could be hidden in the entropy. However, the concentration dependence of $C_p$ in fcc $HfH_x$ appears to be too small to be of significance[33].

The coherent phase transition we observe in thin films, may either be due to the low temperature or the clamping of the film to the substrate. It may be responsible for the extension of the linearity in $\ln(P/P_0)$ to high pressures. Given the fact that in bulk the same linearity is observed as in thin films below the fcc/fct transition, we conclude that this linearity itself is an intrinsic material property. Possibly local hydrogen ordering is the origin of this phenomenon. As our current thermodynamic analysis (see Supplementary Notes) fails to explain the observed behaviour, we conclude that a more detailed study is required of the structural nature of the Hf–H system.

Irrespective of the underlying mechanism, the optical behaviour of Pd-capped Hf is suited to develop a large range optical hydrogen sensor. The fast switching observed in the Pascal range is, to our knowledge, unique[12]. The limited diffusion kinetics at room temperature could be improved by using a local surface plasmon resonance-based architecture[3,4]. Protective coatings such as PTFE are need to be further developed to overcome surface barriers. For realistic applications involving poisonous gasses such as CO, specific coatings need to be developed to prevent poisoning of the Pd surface[30]. To push the sensors to higher pressures other transition metals and in particular Hf-alloys might be considered. To guide the search for such sensing materials a better understanding of the thermodynamic behaviour of these materials is essential.

## Methods

**Sample preparation.** The thin film samples are composed of a 40 nm thick Hafnium layer, capped by a 10 nm thick Pd layer. The layers are deposited on various substrates in 3 μbar of Ar by magnetron sputtering in an ultrahigh vacuum chamber (AJA Int.) with base pressure $10^{-10}$ Pa. The homogeneous thick layers are obtained by a rotating substrate. For the optical and XRD measurements the layers are deposited on $10\times10$ mm² quartz substrates (both sides polished) with a thickness of 1 mm, whereas for NR the layers are deposited on 3″ fused quartz substrates (5 mm thick) with a surface roughness $<4$ Å. The thickness of the layers

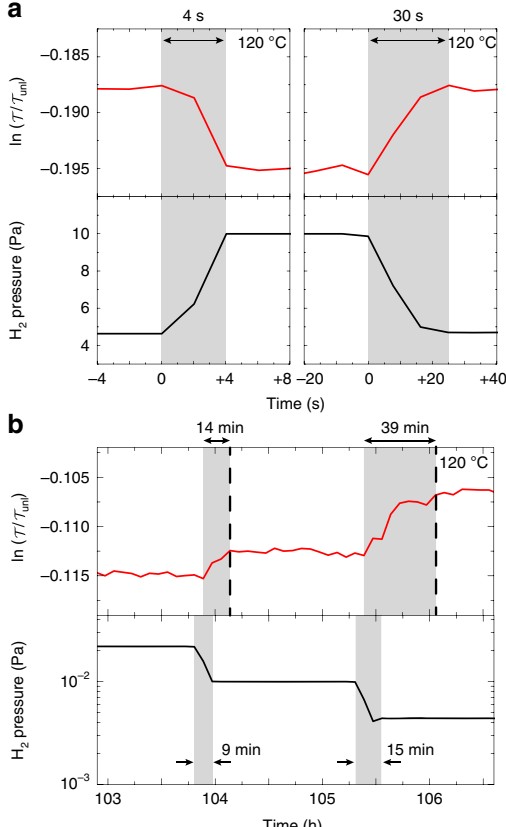

**Figure 7 | Kinetics of HfH$_x$ at different pressures.** The optical response time (in seconds and at 120 °C) to **a** a pressure increase and decrease between 4.7–10 Pa and **b** a pressure decrease around $10^{-2}$ Pa.

is derived from the sputter rate, which is calibrated by stylus profilometry (DEKTAK) on thick samples ($>400$ nm). Typical deposition rates are $1.5 \, \text{Å s}^{-1}$ (125 W direct current (DC)) for Hf and $1.27 \, \text{Å s}^{-1}$ (50 W DC) for Pd. The PTFE coating ($\sim 30$ nm) is produced by radio frequency (RF) sputtering at 3 μbar of Ar[28]. We use a low deposition rate of $0.28 \, \text{Å s}^{-1}$ (70 W) and during deposition typically some fluorine gas is produced.

**Optical experiments.** The optical properties are obtained by means of hydrogenography[13,16,23]. The optical transmission of the film is monitored by means of a three charge-coupled device camera and is averaged over an area of $20 \times 20$ pixels$^2$ (100 pixels$^2$ corresponds to 1 cm$^2$). The various partial hydrogen pressures between $10^{-3}$ and $10^{+4}$ Pa is obtained by using 1 p.p.m. H$_2$/Ar, 0.1% H$_2$/Ar and 4% H$_2$/Ar gas mixtures. The typical gas flow is set to 20 s.c.c.m. for increasing pressure steps and to 200 s.c.c.m. for decreasing pressure steps. The white light source consists of five Philips MR16 MASTER LEDs (10/50 W) with a colour temperature of 4,000 K and a beam angle of 24°. An Al reference sample is used to deal with fluctuations in the LED white light source, except for the stability measurements.

**Structural experiments.** The XRD measurements are performed with a Bruker D8 Advance (Co Kα $\lambda = 1.789$ Å), including an Anton Paar XRK 900 reactor chamber for *in situ* measurements. The NR measurements were performed at Offspec, ISIS, Rutherford Appleton Laboratory, with an incident angle of 8.7 mrad resulting in a Q-range of 0.08–0.8 nm$^{-1}$ with a wave vector transfer resolution of $\Delta Q/Q = 0.05$ (ref. 34). During the NR measurements, the sample was hydrogenated inside a tailor made hydrogenation cell, with controlled pressure, flow and temperature. As a loading gas, a mixture of 99.9% argon and 0.1% H$_2$ was used and a constant flow of 10 s.c.c.m. was maintained. Two sides of the cell are equipped with thin Al windows, to ensure a high transmission for neutrons. After setting the pressure, we waited 5 min before commencing the NR measurements of 20 min per point, to be sure that the sample fully responded to the new pressure set point. The NR measurements were fitted using STAR by minimizing $\chi^2$, to obtain values for the layer thickness, roughness and scattering length density for each layer[35]. The boundaries of the confidence interval of the fitted parameters are subsequently computed by finding the value of the parameter of interest for which an F-test shows that the fit with this value differs 1 s.d. (tail probability of 15.4%) from the value obtained using the best fit. The hydrogen fraction was computed from the fitted parameters by assuming that the number of Hf atoms within a layer remained constant upon hydrogenation (see the supplement of ref. 36).

**DFT calculations.** The DFT calculations are performed in the PAW framework[37] using the HSE06 functional[38] as implemented in the Vienna Abinitio Simulation Package[39]. We use $40 \times 40 \times 40$ gamma centred k-point meshes, $40 \times 40 \times 20$ for the $1 \times 1 \times 2$ fcc supercell for the Hf$_3$H$_2$ structure. For the exchange part of the calculation, the k-mesh is reduced by a factor 5. All computational parameters are set as to reach convergence on the dielectric function within maximally 5% in the visible spectrum. The effect of spin–orbit coupling, tested at Perdew-Burke-Ernzerhof (PBE) level does not affect the visible spectrum significantly.

**Data availability.** The data that support the findings of this study are available from the corresponding author upon reasonable request.

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

## Acknowledgements

We thank Ronald Griessen and Ruud Westerwaal for the fruitful discussions, and Herman Schreuders and Ben Norder for the technical support. We thank the ISIS facility at the Rutherford Appleton Laboratory, Oxfordshire, for the provision of beam time and technical support (RB1610112) and also thank Joshaniel Cooper and Kenneth Elgin for their help and fruitful discussions. Computational resources made available on the Tier-1 supercomputer of the Fédération Wallonie-Bruxelles, infrastructure funded by the Walloon Region under the grant agreement number 1117545 are acknowledged. This work is part of the research of the Stichting voor Fundamenteel Onderzoek der Materie (FOM), which is financially supported by the Nederlandse Organisatie voor Wetenschappelijk Onderzoek (NWO).

## Author contributions

C.B. prepared the samples, performed the optical and the XRD experiments, and analysed the data. L.J.B. and N.-J.S. performed and analysed the data from the NR experiments. C.B. and B.D. wrote and edited the manuscript. M.J.v.S. performed and analysed the DFT calculations. B.D. and A.A.v.W. supervised the project.

## Additional information

**Competing interests:** The authors declare no competing financial interests.

**Publisher's note**: 

