## [Peer Review File · Nature Communications]

PEER REVIEW COMMENTS

Reviewers' Comments:

Reviewer #1 (Remarks to the Author):

The authors rewrote their paper according to publish it in Nature Communications, as suggested by one of the other referees. In the title they restricted the pressure range from 10 to 6 orders of magnitude which is a careful decision. As the information is the same (“the measurement ranges over 6 orders in pressure, the extrapolation suggests that even a sensor ranging over 10 orders in pressure might be feasible!”), I still consider the central message of this manuscript being of high interest. It is expected to lead to research in this field and to the development of a new type of sensors spanning a wide pressure range. Therefore, I support publication in Nature Communications. I am satisfied with the answers the authors gave to my comments.

Just some small comments to the new version:

In Figure 1 the inset marker box in the phase diagram is now at a wrong position – it just be located at the related H-concentrations (the word ‘inset’ is not required, if the authors link the edges of the box to the edges of the enlarged box)

I suggest to remove the ‘!’ in “over 10 orders in pressure might be feasible!”. It is not required to transport the message.

Reviewer #2 (Remarks to the Author):

I have read with interest the revised version of the manuscript on using HfH by Boelsma and coauthors. I have checked carefully whether the authors have incorporated the variety of questions and answers from all three referees in their manuscript, which they did to my satisfaction. Particularly happy makes me the fact that they limited themselves to the HfH system and did not incorporate preliminary Ta data.

The paper definitely gained strength through the incorporation of the additional neutron scattering data (fig. 3).

I believe that the paper is now a very strong one and ready for publication in Nature Communications.

Response to Reviewers:

Reviewer #1 (Remarks to the Author):

Just some small comments to the new version:

In Figure 1 the inset marker box in the phase diagram is now at a wrong position – it just be located at the related H-concentrations (the word ‘inset’ is not required, if the authors link the edges of the box to the edges of the enlarged box)

- *We re-edited this figure. From the x-scale the enlarged section around $x=1.7$ is now evident*

I suggest to remove the ‘!’ in “over 10 orders in pressure might be feasible!”. It is not required to transport the message.

- *We removed the exclamation mark*

Reviewer #2 (Remarks to the Author):

I believe that the paper is now a very strong one and ready for publication in Nature Communications.

- *No changes required*